# Effect of Long-Term Strontium Exposure on the Content of Phytoestrogens and Allantoin in Soybean

**DOI:** 10.3390/ijms19123864

**Published:** 2018-12-04

**Authors:** Sławomir Dresler, Magdalena Wójciak-Kosior, Ireneusz Sowa, Maciej Strzemski, Jan Sawicki, Jozef Kováčik, Tomasz Blicharski

**Affiliations:** 1Department of Plant Physiology, Institute of Biology and Biochemistry, Maria Curie-Skłodowska University, Akademicka 19, 20-033 Lublin, Poland; 2Department of Analytical Chemistry, Medical University of Lublin, Chodźki 4a, 20-093 Lublin, Poland; irek.sowa@gmail.com (I.S.); maciej.strzemski@poczta.onet.pl (M.S.); 91chem91@gmail.com (J.S.); 3Department of Biology, University of Trnava, Priemyselná 4, 918 43 Trnava, Slovak Republic; jozkovacik@yahoo.com; 4Orthopaedics and Rehabilitation Clinic, Medical University Lublin, Chodźki 4a, Lublin 20-093, Poland; tomasz.blicharski@umlub.pl

**Keywords:** *Glycine max* (L.) Merr., strontium, isoflavones, coumestrol, allantoin

## Abstract

Abiotic stress, including metal excess, can modify plant metabolism. Here we investigated the influence of long-term strontium exposure (12 weeks, 0.5–4.0 mM Sr) on the content of phytoestrogens and allantoin as well as the mineral composition in soybean. Seven phytoestrogens were identified in the soybean: daidzin, glycitin, genistin, malonyldaidzin, malonylgenistin, daidzein, and coumestrol. The results showed that both malonyldaidzin and malonylgenistin were dominant phytoestrogens; however, the roots contained a relatively high amount of daidzein. It was found that strontium reduced the phytoestrogen content and decreased the antioxidant capacity. Strontium evoked depletion of the sum of all phytoestrogens by 40–70% in the leaves, 25–50% in the stems and in the seeds, depending on the strontium concentration. In the roots, 0.5 and 4.0 mM of strontium decreased the total phytoestrogen content by 25 and 55%, respectively, while 2.0 mM of strontium did not exert an effect on their accumulation. On the other hand, strontium ions induced allantoin accumulation mainly in the roots. Strontium was preferentially accumulated in the leaves, with a slight impact on macro- and micro-nutrients. Our research showed strontium-secondary metabolites interaction in the soybean, which can be useful for obtaining a natural pharmaceutical product containing both strontium and phytoestrogens for remediation of postmenopausal osteoporosis.

## 1. Introduction

Plant biologically active compounds are important for pharmaceutical use or for obtaining nutraceutical products such as functional food and dietary supplements. Accumulation and biosynthesis of such metabolites in plants depend on many factors, including biotic and abiotic stress. Generally, as a consequence of stress factors such as pathogen activity, UV-radiation, high or low temperature, salinity, or exposure to metals, the potency of enzymes of the phenylpropanoid pathway are altered, which leads to changes in the accumulation of secondary metabolites [1,2,3]. Isoflavones are a group of active compounds accumulating in soybean in a genotype-dependent manner [4,5]. Moreover, numerous reports have shown that accumulation of these compounds is highly altered in response to environmental stresses [6,7,8,9,10]. It was observed that the concentration of isoflavones increased in plants in response to various stresses, e.g., biological—pathogen attack [9]; physical—solar UV-B radiation [11], sonication and vacuum infiltration [7], pulsed electric field [6], low temperature [12], or chemical elicitors such as methyl jasmonate [7]. On the other hand, Chennupati et al. [4] pointed out that high temperature can significantly decrease the accumulation of these compounds in soybean pods and seeds. Isoflavones, together with coumestans and prenylflavonoids, belongs to the phytoestrogens, i.e., a diverse group of nonsteroidal plant metabolites [13]. Phytoestrogens occur in many plant species such as the red clover (*Trifolium pratense* L.), kudzu (*Pueraria lobata*, Willd.), and soybean (*Glycine max* L.), which is considered to be the most valuable source of these compounds [14,15]. Soybean isoflavones have exhibited many therapeutic effects, including prevention of cancers or protection against cardiovascular diseases; moreover, they have antioxidant, antibiotic, and anti-inflammatory properties. Additionally, phytoestrogens have exhibited protective action against bone mass lost and micro-architectural deterioration of bone tissues. Since osteoporosis is associated with a reduced level of estrogen, it has been suggested that the mechanism of the phytoestrogen effect on bone health is based on mimicking estrogen, which leads to reduced osteoclastic bone resorption and stimulation of bone formation by osteoblastic cells [13,16]. A similar therapeutic impact on bone density is exerted by strontium, which is used as a strontium ranelate in therapy of human osteoporosis. Based on ample evidence of therapeutic implications of strontium in bone healing and bone fracture repair, Saidak and Marie [17] indicate that this element should be supplemented in osteoporosis therapy and prevention. 

Previous studies indicate that strontium can be efficiently accumulated by plants. Sowa et al. [18] pointed out that young soybean seedlings can uptake and transfer strontium to aboveground part of plants very efficiently. They found that the concentration of strontium in shoots is approximately 7–9 times higher than in roots. Additionally, it was shown that the strontium content in plants tissues is strongly positively correlated with its concentration in the growth medium (*R* > 0.98). Another paper [10] indicated that the presence of strontium in the nutrient medium can be used to functionalize soybean plants. The authors pointed out that exposure of soybean seedlings to strontium for 14 days significantly increased the level of phytoestrogens in the plants.

The present study investigated the long-term strontium impact on some secondary metabolites and minerals in soybean plants. In particular, we intended to check whether (i) different long-term strontium concentrations in hydroponic culture altered the accumulation of phytoestrogens and allantoin in different soybean organs (leaves, shoots, seeds, roots); (ii) the antioxidant capacity of soybean extracts were changed by strontium ions; (iii) the long-term strontium exposure led to biofortification of soybean plants with this element; and (iv) the presence of strontium in the growth medium affected the content of some macro and microelements in soybean organs. 

## 2. Results and Discussion

### 2.1. Impact of Strontium Ions on Soybean Growth

The plant shoot biomass was not affected by the presence of strontium in the nutrient medium, while the 4.0 mM dose of strontium significantly increased roots biomass, compared with the control plants (Table 1). The stable forms of strontium are not considered especially toxic to plants, while their adverse effect on plant development and growth is very often combined with its negative impact on the uptake of some nutrients, especially calcium [19]. Previously, it was shown that 1.5 mM of strontium stimulated soybean seedlings [18]. An increase in biomass production was also observed in maize plants treated with 0.1 and 1.0 mM of strontium [20]. On the other hand, a previous study indicated that a higher concentration of this element (over 2 mM) inhibited the growth of soybean seedlings [18]. Moreover, the toxic effect of strontium on growth both in higher plants [21,22] and algae [23] were observed in previous hydroponic experiments. 

### 2.2. Changes in Secondary Metobolites and Antioxidant Capacity in Response to Strontium 

Based on comparison of retention times and UV-VIS spectra of individual peaks (Appendix A) between the plant samples and standards, seven phytoestrogens were identified in soybean organs; additionally, a diureide of glyoxylic acid–allantoin was determined (Appendix A). As shown in Figure 1, both malonylglucoside compounds—malonyldaidzin and malonylgenistin—were the dominant isoflavones present in the soybean tissues; in turn, the roots contained a high concentration of an aglycone—daidzein. This aglycone was not detected in the leaves and seeds. Moreover, glycitin was not detected in the stems and roots, while coumestrol and allantoin were not present in the seeds (Figure 1).

For clarity of results, heat maps were created for all standardized data (Figure 2). The parameters determined for each individual are represented by colors (dark blue—very low value; dark red—very high value). Four values for the individual treatments are shown. 

It was found that, except for glycitin, the concentrations of all the studied phytoestrogens significantly decreased in the strontium-treated leaves (Figure 1 and Figure 2). A clear inhibition effect of strontium on daidzin accumulation was observed in the stems and seeds. The content of malonylglucosides in the stems and seeds differed in relation to the strontium concentration. It was noted that 0.5 and 4.0 mM of strontium significantly reduced both malonyldaidzin and malonylgenistin amounts compared with the control plants. A similar negative effect of strontium on the genistin content in the stems and seeds was shown as well. On the other hand, the 0.5 mM dose of strontium increased the daidzein content in the stems by 140%, compared to the control. In general, strontium did not influence the total phenolic content (TPC) and soluble flavonols (Table 2); however, the stems exhibited lower TPC after the treatment with 4.0 mM of strontium, in comparison with control. The 4.0 mM dose of strontium also decreased antioxidant capacity in the leaves and stems, while the radical scavenging activity in the seeds and roots was similar between the strontium treatments. 

The negative impact of strontium on the phytoestrogen concentration is surprising, since a previous paper [10] indicated that strontium induced phytoestrogen accumulation in soybean during 14-day strontium exposure. In this study, the increasing concentration of strontium in the nutrient medium had a positive effect on the phytoestrogen content in the soybean shoots and roots, reaching the maximal level at 2.0 mM of Sr. The authors found that the accumulation of daidzein, genistein, and coumestrol increased 4.2-, 2.0-, and 3.9-fold, respectively, in the shoots and 2.7-, 4.1-, and 2.0-fold, respectively, in the roots, compared to plants cultivated without strontium addition. The absence of a positive effect of strontium on the phytoestrogen content observed in the present work is probably a result of the long-term strontium exposure and the cumulative effect of this element. In the aforementioned study, Wójciak-Kosior et. al [10] showed an increasing effect of strontium on isoflavones but only up to 2.0 mM. A further increase in the dose of this element in the medium caused reduction of the content of these compounds. Another paper indicated that accumulation of flavonoids in response to metal stress significantly depends on the duration of exposure to stress factors [3]. It was demonstrated that chronic heavy metal stress significantly decreased the concentration of flavonoids, while acute stress in a short-term experiment increased rutin and total flavonoid content [3]. 

On the other hand, it was found that the accumulation of allantoin significantly increased under the strontium excess (Figure 1). This phenomenon was especially evident in the roots, in which accumulation of allantoin increased 4.4, 3.2, and 4.0-fold in the roots treated with 0.5, 2.0 and 4.0 mM of strontium, respectively, compared to the control. Allantoin is an important nitrogenous compound present in both nodulated and non-nodulated plants [24,25]. An elevated concentration of allantoin in response to water deficit [26], salinity [27], irradiance [28], or excess of heavy metals [3,29] has been demonstrated previously. Additionally, there is evidence that allantoin contributes to protection against environmental stress [30]. It is proposed that the role of this ureide is based on activation of antioxidant enzymes [31].

### 2.3. Strontium and Some Macro- and Microlement Concentrations

Biofortification of crops with elements with nutritional or therapeutic value can be useful for obtaining functional plant products [18]. Our results indicated that accumulation of strontium in the soybean strongly depended on its concentration in the nutrient medium (Figure 2 and Figure 3). The highest accumulation of strontium (above 17 mg·g^−1^ ADW of leaves) was observed in the 4.0 Sr mM treatment and, in comparison with other plants species [32], the results obtained indicated that soybean has a great ability to accumulate Sr. Kabata-Pendias and Mukherjee [32] showed that the content of strontium in plants is highly variable and ranges from below 1 to 10,000 mg·kg^−1^ DW; however, the most common accumulation of this element in different crops ranges from ca. 10 to 1500 mg·kg^−1^ DW. Moreover, the authors pointed out that legume herbages, contrary to grains, fruits, or potato tubers, are species that accumulated the highest amounts of strontium. 

It was found that strontium was easily translocated to the leaves and, generally, its accumulation in plant organs decreased in the following order: leaves > stems > roots > seeds. The low strontium content in the seeds was expected, since the transfer of non-essential metals to generative organs is usually limited [33]. The high ability of soybean to translocate strontium from roots to shoots was described with the transfer factors (TF) expressed as a ratio between the content of strontium in the roots and shoots. The TFs calculated for strontium were 9.3, 2.6, and 3.5 for plants treated with 0.5, 2.0, and 4.0 mM of strontium, respectively. These data correspond with a previous paper, which reported that strontium was easily transferred to the aboveground part of soybean [18]. The tendency toward translocation of strontium to leaves was also observed in *Sorghum bicolor* cultivated in strontium-contaminated soil [34] and in plants growing in a mining area [35]. Burger and Lichtscheidl [19] pointed out that strontium is very easily translocated to the above parts of various plant species with a translocation factor higher than 1. Previous studies have shown that, besides leaves, strontium was accumulated in the stem of *Oryza sativa* [36], leaf and stem hairs [37], or cortex of conifer trees [38]. 

Moreover, the comparison of the present results of the strontium concentration in the shoots (leaves and stems) with those from a previous study [18] indicated that accumulation of this element strongly depended on the duration of exposure. It was found previously that young soybean plants subjected to 14-day strontium exposure accumulated 17- and 7-fold lower strontium levels [18] compared with results obtained from a long-term experiment (84-days exposure to strontium) presented in this paper. With its high physical and chemical similarity to calcium, strontium resembles the latter element in terms of its uptake and distribution in the plant [32]. While calcium is an essential element for plant growth and development, strontium is considered to be an element without any identified physiological role in plants; however, strontium can be passively taken up and replace calcium [39].

As shown in Figure 3b, strontium significantly influenced calcium accumulation in the leaves and roots. Since both these elements are taken up by plants in a similar way via calcium channels, we can expect some competition and antagonism in the strontium and calcium uptake by plants [39]. This fact may explain the negative effect of 0.5 an 4.0 mM of strontium on the calcium content in the roots observed in the studied soybean plants. A decrease in the calcium concentration in plants cultivated at excess of strontium was observed previously as well [20,40]. Surprisingly, the excess of strontium in the nutrient medium had a positive impact on the level of calcium in the leaves, potassium in the stems and roots, and magnesium in the leaves and roots (Figure 3). Moreover, the plant exposure to strontium increased the accumulation of nickel and copper in the stems and roots and manganese in the stems (Appendix A). Although it was shown previously that strontium reduces accumulation of some elements in plants [20,40], Gupta et al. (and references therein) [39] indicated a positive correlation between the concentrations of strontium and other elements. Partly contradictory results were obtained in previous studies [20,40], indicating a complex process of nutrient uptake from a medium supplemented with strontium, depending on many factors, including the type of species, time and concentration of strontium exposure, or balance between nutrients in the growth medium. 

### 2.4. Principal Component Analysis

The results of principal component analysis (PCA) for different soybean organs are shown in Figure 4. The PCA of the secondary metabolites, antioxidant capacity, and macroelement compositions showed clear separation between the control (0 mM of strontium) and the strontium-treated plants. However, the strontium-treated individuals generally did not group according to the concentration of strontium in the medium.

It was noted that the first PCs and the second PCs explained 46–52% and 16–19% of the total variance, respectively, depending on plant organs. Generally, the first PCs (in the analysis of all plant organs) were strongly correlated with phytoestrogen accumulation, TPC, soluble flavonols, and antioxidant capacity. Additionally, this component was determined by strontium, potassium, magnesium, and partially allantoin; however, these variables are negatively correlated with the secondary metabolite parameters mentioned above. In the leaves, stems, and seeds, mostly the first PCs facilitated separation of the control plants (individuals with a high concentration of phytoestrogens, TPC, and antioxidant capacity) from the strontium-treated individuals (high content of strontium, potassium, magnesium, and allantoin). In the roots, three of the four control objects were strongly distinguished from the other samples by PC2, which showed a high correlation with calcium.

## 3. Materials and Methods 

### 3.1. Plant Material

The soybean seeds (*Glycine max* L. Merr. subsp. *soja*) were germinated on wet filter paper and 3-day old seedlings were transferred into pots filled with Hoagland’s nutrient (five plants per pot). After 2 weeks, the nutrient solution was supplemented with 0-control, 0.5, 2.0, or 4.0 mM Sr as Sr(NO_3_)_2_. Eight pots (40 plants) were used in each treatment. The plants were cultivated for 12 weeks until complete formation of pods. The nutrient solution was continuously aerated and changed every 10 days, while its losses were supplemented daily with distilled H_2_O. The plants were cultivated in controlled conditions in a growth chamber equipped with red and blue light-emitting diodes under a 16/8 h (day/night) photoperiod at 25/17 °C (day/night). After harvest, the roots of the plants were washed with distilled water and carefully dried with filter paper. The plants were weighed (shoots and roots), divided into leaves, stems, seeds, and roots and dried at room temperature over 4 days. Afterwards, the samples combined from 10 plants (2 pots); leaves, stems, seeds, and roots separately were divided into two parts. One part of the sample was milled into powder and secondary metabolites were extracted, while the other part of the sample was dried at 80°C for metal analysis. 

### 3.2. Sample Preparation

The air-dried samples were pulverized, and 0.5 g of plant material was extracted with 3 mL of 80% methanol in an ultrasonic bath for 30 min. The procedure was repeated two times with a fresh portion of 1 mL of 80% methanol. The combined extracts were filled to 5 mL of the final concentration and filtered through 0.22 µm nylon filters. Phytoestrogens, allantoin, antioxidant capacity, total phenolics, and soluble flavonols were determined in these extracts. 

### 3.3. HPLC Analysis of Phytoestrogens 

HPLC determination of phytoestrogens was performed on a VWR Hitachi Chromaster 600 chromatograph with a spectrophotometric detector DAD according to the method described previously [33]. The separation was carried out using a C18 reversed-phase column LiChrospher 100 (Merck, Darmstadt, Germany) (25 cm × 4.0 mm i.d., 5 µm particle size) at a temperature of 30 °C. The mobile phase consisted of acetonitrile—solvent A and water with 0.025% of trifluoroacetic acid—solvent B. The gradient program was as follows: A 15%, B 85% for 0–5 min; A 20%, B 80% for 5–15 min; and A 25%, B 75% for 15–45 min. The flow rate of the eluent was set at 1.5 ml min^−1^. The data were collected in a wavelength range from 200 to 400 nm. The identity of the phytoestrogens was established by comparison of retention times and UV-Vis spectra with those obtained for corresponding standards purchased in Sigma Aldrich (St. Louis, MO, USA). 

### 3.4. Allantoin Determination Using Capillary Electrophoresis

The allantoin content was measured with a method described previously [41] with some further modification [24]. An Agilent 7100 Capillary Electrohoresis set coupled with a diode-array detector (UV-Vis/DAD, 190–600 nm; Agilent Technologies, Santa Clara, CA, USA) was used for separation and determination of allantoin. The analysis was carried out in fused silica 50 µm i.d., capillaries with a total length of 64.5 cm. The background electrolyte with pH 9.2 consisted of a 50 mM borate solution. The quantitative analysis of allantoin was performed at λ = 192 nm. Allantoin was identified based on comparison of the retention time and absorption spectrum similarity in a 190–400 nm range with allantoin standards purchased in Sigma-Aldrich (St. Louis, MO, USA). 

### 3.5. Determination of Total Phenolics, Soluble Flavonols, and Antioxidant Capacity 

The extracts used for determination of phytoestrogens and allantoin were used for evaluation of TPC and soluble flavonols and measurement of antioxidant capacity. The TPC expressed as mg of gallic acid equivalents per gram of air-dry weight of plants was determined using the Folin-Ciocalteau reagent as described previously [42]. The concentration of total soluble flavonols expressed as mg of rutin equivalent per gram of air-dry weight was determined based on formation of a flavonol-aluminum complex [43]. The antioxidant capacity expressed as mg of Trolox equivalent per gram of air-dry weight was measured using free radical 2-azino-bis-3-ethylbenzthiazoline-6-sulphonic acid (ABTS) and 2,2-diphenyl-1-picrylhydrazyl (DPPH) [3].

### 3.6. Determination of Strontium and Macro- and Microelements 

The leaves, stems, and roots were dried separately at 80 °C and wet digested in a mixture of HNO_3_:H_2_O (2:8 *v*/*v*) in the microwave digestion apparatus (TOPwave, Analytick Jena AG, Jena, Germany). The concentrations of elements were analyzed using a ContrAA 700 high-resolution continuum source graphite tube atomic absorption spectrometer (Analytik Jena AG, Jena, Germany). The working parameters were specified in our previous paper [18].

### 3.7. Statistical Analysis

The statistical analysis was performed using Statistica ver. 13.3 (TIBCO Software Inc. 2017). One-way ANOVA (analysis of variance) was used for estimation of the significant effect of strontium on the investigated parameters. The differences between the treatments were determined with Tukey’s honest significant difference test at the 0.05 probability level. The PCAs were performed separately for each organ; they were based on the concentration of phytoestrogens, allantoin accumulation, TPC, total soluble flavonol content, antioxidant capacity, strontium, and some macronutrient compositions. The heat maps were created based on standardized data using Microsoft Excel (2010) software. 

## 4. Conclusions

Since strontium exhibits high physical and chemical similarity to calcium, which is an essential element for plants, it is not regarded as a typical stress factor. However, the present investigation indicated that excess of strontium in the nutrient medium can disturb the content of secondary metabolites and the composition of mineral nutrients. The study showed that strontium generally reduced the accumulation of phytoestrogens in soybean tissues; on the other hand, plants exposed to this element contained significantly more allantoin than the control plants. Additionally, our results indicated a cumulative effect of strontium accumulation in the long-term experiment. The high level of strontium translocation to leaves resulted in high accumulation of this element in the aboveground parts of plants. Moreover, our investigation showed that biofortification of soybean with strontium may be useful to functionalize this plant and obtain functional food. 

## Figures and Tables

**Figure 1 ijms-19-03864-f001:**
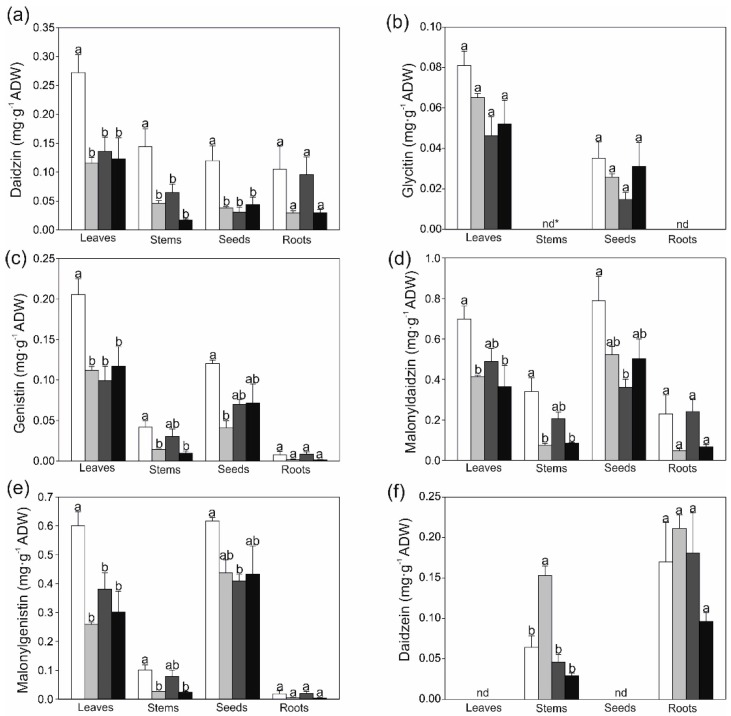
Effect of various strontium concentrations on the content of phytoestrogens and allantoin in air-dry weight (ADW) of different soybean organs; (**a**) daidzin; (**b**) glycitin; (**c**) genistin; (**d**) malonyldaidzin; (**e**) malonylgenistin; (**f**) daidzein; (**g**) coumestrol; (**h**) allantoin. Data are means ± SE (*n* = 4); * nd-not detected. Values followed by the same letters within the same plants’ organ are not significantly different (*p* < 0.05, Tukey’s test).

**Figure 2 ijms-19-03864-f002:**
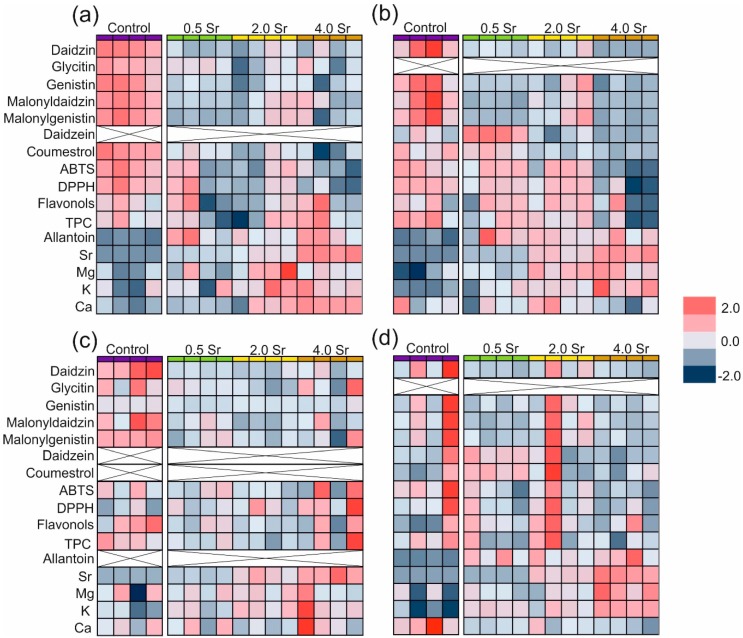
Heat map visualization of changes in the abundance of particular features shown in the rows for individual plant organs treated with different Sr concentrations (column). The colors range from dark blue (low abundance) to deep red (high abundance); (**a**) leaves; (**b**) stems; (**c**) seeds; (**d**) roots.

**Figure 3 ijms-19-03864-f003:**
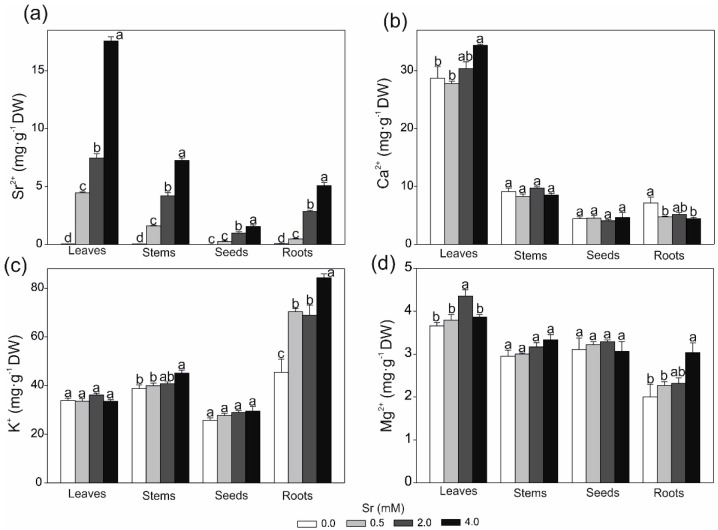
Effect of various strontium concentrations on the content of (**a**) strontium; (**b**) calcium; (**c**) potassium; and (**d**) magnesium in different soybean organs. Data are means ± SE (*n* = 4). Values followed by the same letters within the same plants’ organs are not significantly different (*p* < 0.05, Tukey’s test).

**Figure 4 ijms-19-03864-f004:**
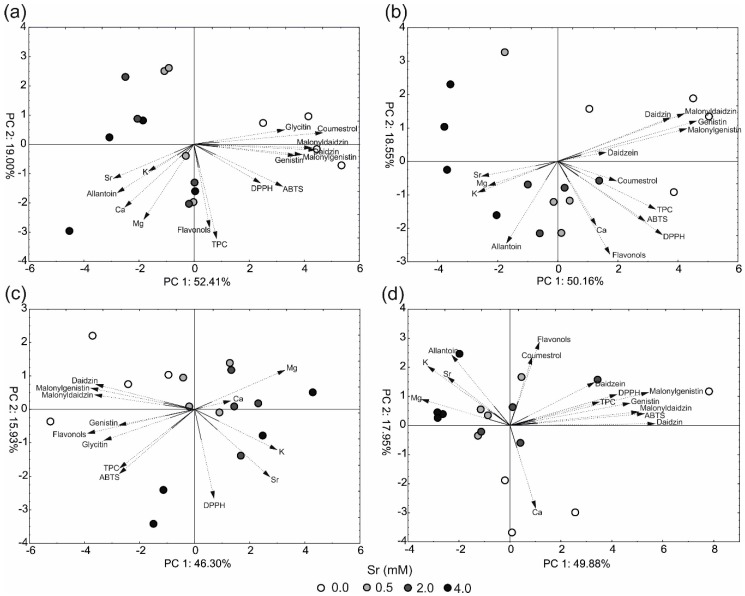
Scatter plot of principal component analysis of selected secondary metabolites, antioxidant capacity, and elements in the soybean organs; (**a**) leaves; (**b**) stems; (**c**) seeds; and (**d**) roots. The length of the dark lines shows a correlation between original data and principal component axes.

**Table 1 ijms-19-03864-t001:** Effect of the different strontium concentrations (mM) in the nutrient medium on the fresh biomass of soybean. Data are means ± SE (*n* = 35–40); the values followed by the same letters are not significantly different (*p* < 0.05, Tukey’s test).

	0.0 Sr	0.5 Sr	2.0 Sr	4.0 Sr
Shoot biomass (g per plants)	4.40 ± 0.42 a	3.74 ± 0.13 a	4.04 ± 0.33 a	3.88 ± 0.33 a
Root biomass (g per plants)	2.59 ± 0.11 b	2.34 ± 0.14 b	3.03 ± 0.19 ab	3.37 ± 0.31 a

**Table 2 ijms-19-03864-t002:** Effect of various Sr concentrations (mM) in the nutrient medium on the total phenolic content (TPC), soluble flavonolos, and antioxidant capacity determined using 2-azino-bis-3-ethylbenzthiazoline-6-sulphonic acid (ABTS) and 2,2-diphenyl-1-picrylhydrazyl (DPPH) in the *Glycine max* plants. Units: mg TE g^−1^ ADW (for antioxidant capacity), mg g^−1^ ADW (for TPC), mg g^−1^ ADW (for soluble flavonols). Data are means ± SE (*n* = 4). Different letters denote significant difference at the 0.05 level of Tukey’s test between the Sr treatments within the same plants organ.

	Total Phenolics	Soluble Flavonols	Antioxidant Capacity	
			ABTS	DPPH
**Leaves**				
0.0 Sr	8.18 ± 0.74 a	4.48 ± 0.34 a	21.1 ± 0.77 a	1.666 ± 0.113 a
0.5 Sr	7.67 ± 0.88 a	4.52 ± 1.25 a	17.6 ± 1.20 ab	1.509 ± 0.157 ab
2.0 Sr	8.01 ± 0.83 a	4.30 ± 0.25 a	16.7 ± 0.89 ab	1.314 ± 0.081 ab
4.0 Sr	9.27 ± 0.73 a	4.88 ± 0.46 a	15.3 ± 1.05 b	1.086 ± 0.119 b
**Stems**				
0.0 Sr	3.14 ± 0.22 a	0.64 ± 0.05 a	2.5 ± 0.09 a	0.635 ± 0.009 a
0.5 Sr	2.13 ± 0.40 ab	0.60 ± 0.09 a	2.1 ± 0.22 ab	0.481 ± 0.069 ab
2.0 Sr	2.35 ± 0.20 ab	0.65 ± 0.04 a	2.2 ± 0.10 a	0.565 ± 0.022 a
4.0 Sr	1.59 ± 0.24 b	0.53 ± 0.07 a	1.6 ± 0.06 b	0.403 ± 0.037 b
**Seeds**				
0.0 Sr	1.96 ± 0.06 a	0.21 ± 0.01 a	4.7 ± 0.13 a	0.417 ± 0.031 a
0.5 Sr	1.77 ± 0.04 a	0.19 ± 0.01 a	4.6 ± 0.12 a	0.455 ± 0.021 a
2.0 Sr	1.54 ± 0.13 a	0.17 ± 0.00 a	4.5 ± 0.06 a	0.498 ± 0.032 a
4.0 Sr	1.78 ± 0.21 a	0.18 ± 0.02 a	4.8 ± 0.30 a	0.537 ± 0.047 a
**Roots**				
0.0 Sr	1.82 ± 0.22 a	0.39 ± 0.08 a	1.7 ± 0.19 a	0.438 ± 0.067 a
0.5 Sr	1.50 ± 0.28 a	0.53 ± 0.06 a	1.3 ± 0.09 ab	0.383 ± 0.035 a
2.0 Sr	2.03 ± 0.34 a	0.53 ± 0.07 a	1.6 ± 0.12 ab	0.459 ± 0.049 a
4.0 Sr	1.28 ± 0.14 a	0.45 ± 0.06 a	1.2 ± 0.04 b	0.321 ± 0.020 a

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
