# Peer review of "Effect of Long-Term Strontium Exposure on the Content of Phytoestrogens and Allantoin in Soybean"

_ijms, 2018, doi:10.3390/ijms19123864_

Round 1
Reviewer 1 Report
The authors of this manuscript investigate the metabolic impact of long term strontium exposure on soybean plants, assessing the hypothesized capacity of strontium treatments in soybean bio-fortification and antioxidant capacity enhancement. The data indicate a strong negative, cumulative effect of long-term treatments on the accumulation of phytoestrogens in soybean , and show an enhanced intra-plant translocation of strontium to above ground organs, and a good potential for bio-fortification efforts, for later nutritional and medicinal applications.
The study is interesting and relevant and is well written. I only have a couple of minor questions/comments:
- The authors might want to consider slightly expanding the discussion of the obtained data. Also, while their work did not include physiological measurements, it would be worth discussing if prior studies have assessed the morpho-physiological impacts of increasing Sr amounts in plants, potential physiological or toxic concentration values, toxicity symptoms, whether there is a hypothesized or known mechanism of Sr accumulation in plants, are there species which are hyper-accumulators and whether soybean is one of them, etc.
- in Table 1 and 2 include in the heading/first column that the values are actually mM concentrations.
- the authors air dried the plant materials used for the analyses. Please include information on how long the materials were dried.
Author Response
Dear Reviewer,
Thank you very much for the fast and precise comments. They have all been considered and incorporated as indicated in detail below. All changes are marked in the red color in the revised text.
We hope that the modifications done in the text will be acceptable for you.
Yours sincerely,
S. Dresler, corresponding author
- The authors might want to consider slightly expanding the discussion of the obtained data. Also, while their work did not include physiological measurements, it would be worth discussing if prior studies have assessed the morpho-physiological impacts of increasing Sr amounts in plants, potential physiological or toxic concentration values, toxicity symptoms, whether there is a hypothesized or known mechanism of Sr accumulation in plants, are there species which are hyper-accumulators and whether soybean is one of them, etc.
REPLY: thank you for this good suggestion. We monitored only biomass during the experiment and it is true that some physiological measurements could be suitable, but as we did not find any impact of Sr on (for example) the content of photosynthetic pigments or morphometry parameters (for example leaf area), the data were limited only to the biomass parameters. It would of course be interesting to study the Sr accumulation potential of various plant species in our future works. The discussion has been slightly extended.
- in Table 1 and 2 include in the heading/first column that the values are actually mM concentrations.
REPLY: the caption of second table has been corrected
- the authors air dried the plant materials used for the analyses. Please include information on how long the materials were dried.
REPLY: the necessary information has been included into the Method section
Reviewer 2 Report
The paper is easy to read and the results are presented clearly. My only concern is regarding the Materials and Methods, viz. design and sampling?
Given the genetic significance of phytoestrogen content I was surprised there was no comment on that in the Introduction? You should insert a statement declaring which cultivar of G max you cultivated.
Design: Presumably 5 non-nodulated seedlings were cultivated in each of 8 pots per Sr treatment: viz. 4 rates x 8 reps. For phytoestrogen and mineral data (Fig. 2 & 3), n=4. For phenolics, flavonols, antiox. activity (Fig 1, Table 2), n=6. For your study to be reproduced the reader needs to know how your analytical sample was collected. State clearly what was done. Were all plants processed? If all 'replicates' were not used how were those that were used selected?
These details should be clarified.
The variability of phytoestrogen data is already described in Fig 2. Why then show 4 data 'points' on the heat map? Would simple line plots or regression analysis be a more useful way of communicating what the heat map attempts?
Line 299: delete "of"?
Author Response
Dear Reviewer,
Thank you very much for the fast and precise comments. They have all been considered and incorporated as indicated in detail below. All changes are marked in the red color in the revised text.
We hope that the modifications done in the text will be acceptable for you.
Yours sincerely,
S. Dresler, corresponding author
Given the genetic significance of phytoestrogen content. I was surprised there was no comment on that in the Introduction? You should insert a statement declaring which cultivar of G max you cultivated.
REPLY: A short comment and references reporting the genetic significance of the phytoestrogen content have been given in the Introduction. Additionally, information about the cultivated taxon has been included.
Design: Presumably 5 non-nodulated seedlings were cultivated in each of 8 pots per Sr treatment: viz. 4 rates x 8 reps. For phytoestrogen and mineral data (Fig. 2 & 3), n=4. For phenolics, flavonols, antiox. activity (Fig 1, Table 2), n=6. For your study to be reproduced the reader needs to know how your analytical sample was collected. State clearly what was done. Were all plants processed? If all 'replicates' were not used how were those that were used selected? These details should be clarified.
REPLY: thank you very much for this suggestion. The samples for the analysis of SMs or metals were prepared in the following way:
“the samples combined from 10 plants (2 pots) –leaves, stems, seeds, and roots separately were divided into two parts. One part of the sample was milled into powder and secondary metabolites were extracted, while the other part of the sample was dried at 80°C for metal analysis.”
We are sorry but we made an error during preparation of the manuscript: it should be n=4 (not n=6) in all cases. The error has been corrected. Moreover, the editorial suggestions have been incorporated into the text of the manuscript.
The variability of phytoestrogen data is already described in Fig 2. Why then show 4 data 'points' on the heat map? Would simple line plots or regression analysis be a more useful way of communicating what the heat map attempts?
REPLY: the heat map is one of the methods to show the obtained results. Since we measured many different parameters, including the content of phytoestrogens and some metals, in our opinion the use of the heat maps was appropriate to show the results obtained in an easy, fast, and readable way.
Line 299: delete "of"?
REPLY: the “of” has been deleted